# A Survey of the Status and Challenges of Green Building Development in Various Countries

**Yinqi Zhang [1], He Wang [1], Weijun Gao [1,2,*], Fan Wang [3], Nan Zhou [4], Daniel M. Kammen [5] and Xiaoyu Ying [6]**

1   School of Environmental Engineering, the University of Kitakyushu, Kitakyushu 8080135, Japan;
    yinqiz90@gmail.com (Y.Z.); wangheawork@gmail.com (H.W.)
2   iSMART, Qingdao University of Technology, Qingdao 266033, China
3   Royal Academy of Engineering Centre of Excellence in Sustainable Development Building Design, Heriot
    Watt University, Edinburgh EH14 4AS, UK; fan.wang@hw.ac.uk
4   Lawrence Berkeley National Laboratory, Berkeley CA 94720, USA; NZhou@lbl.gov
5   Energy and Resources Group (ERG); The University of California at Berkeley, Berkeley CA 94720, USA;
    kammen@berkeley.edu
6   Department of Engineering, Zhejiang University City College, Hangzhou 310015, China;
    yingxiaoyu@zucc.edu.cn
*   Correspondence: gaoweijun@me.com

**Abstract:** Since the energy crisis in the 1960s, crucial research and activities were spurred to improve energy efficiency and decrease environmental pollution. To deal with the various problems the construction industry are facing, the concept of green buildings (GBs) has been gradually shaped and put forward all over the world, and green building rating systems (GBRSs) have been developed. The concept of GBs covers a wide range of elements, and its definition is constantly updated as the construction industry develops. This paper compares the development of backgrounds and statuses of green building development in various countries. It also presents an overview of the green building development situation within these countries, summarizing two influences for GB development: one external and the other internal. External factors include GB development policy support, economic benefits, and certification schemes. Internal factors are the development and application of GB technology, the level of building management, and how users interact with the GB technology. Currently, 49 worldwide green building standards and application have been sorted out, including 18 standard expert appraisal systems. Moreover, it discusses the research results and lessons learned from green building projects in different countries and summarizes their achievements and challenges. To correctly understand and use green building technology, it is essential to improve the policy and incentive system, improve the professional quality and technical ability of employees and accredited consultants, constantly develop and update the evaluation system, strengthen technological innovation, and integrate design and management. This paper aims to draw a clear roadmap for national standard development, policy formulation, and construction design companies, provide solutions to remove the obstacles, and suggest research direction for future studies.

**Keywords:** green buildings; sustainable building; green building technologies; green building rating systems

## 1. Introduction

Today's global issues like climate change, energy shortages, increasing environmental pollution, rising population, and rapid urbanization present tremendous challenges to the sustainable development of human society [1]. NASA reports that the global average temperature has increased

by 1.8 °F since 1880 [2]. The rise in global average temperature is expected to be about 4.5 °F by 2050 from the $CO_2$ increase alone [3]. The world's carbon dioxide ($CO_2$) emissions from energy-related consumption will increase from 32.3 billion metric tons in 2012 to 43.4 billion metric tons in 2040 [4]. Meanwhile, the growing population continues to place a heavier burden on the environment. According to World Population Prospects 2017, during the 13 years from 2005, the world's population had added about one billion newborns, and world population would reach 9.8 billion in 2050 [5]. This increasing population and galloping urbanization are accelerating the demand for energy [1] that will reach 900 EJ primary energy use in 2050 [6].

Among those various causes of these problems, the building construction industry has been criticized as being a leading exploiter of a large proportion of primary energy and natural resources [7]. Globally, the industry has made a significant impact on our resources, environment, society, economy, and human health. It consumes 30% of global resources, 15% of global freshwater withdraws, one-fourth of wood harvested, and nearly half of raw materials used [7]. The $CO_2$ released from the energy used to produce tiles, glass, concrete, and other construction materials is more than those of industry and transport [1]. The building sector generates 30% of the world's greenhouse gases [8] and 40%–50% of water pollution to the environment [1]. Additionally, it contributes 40% of the total solid waste in developed countries [9]. To address these issues, the construction of green buildings (GB) focuses on improving building energy efficiency and alleviating construction's negative impacts on the environment and resources [10]. It can integrate strategies from all building life cycle stages, including siting, design, construction, operation, maintenance, renovation, and deconstruction to reduce the negative impacts on energy, water, materials, and other natural resources. It can also decrease environmental pollution from waste, air and water pollution, indoor pollution, heat islands, stormwater runoff, noise, and more [11]. The introduction and implementation of GBs have indeed achieved reduction in energy consumption and $CO_2$ emission and improvement in water management in many projects. At least in their design proposals, the designers demonstrate their intentions to follow GBs guidance to achieve the best outcome.

Although GB certification programs and the square footage they cover are increasing each year, they are still far from the total floor area of the huge building market. This is partly due to the many restrictions on the promotion of GBs. Also, although extensive research has examined various aspects of GBs, there has been a lack of systematic review of the state of the art and future tendencies from around the world, including developing countries. This paper presents a critical overview of GB development status in various countries and related studies by discussing the research results produced by GB technology implementation, looking at both external and internal factors. The goals of this paper are to draw a clear roadmap for national standard development, policy formulation, and construction design companies, offer guidance for overcoming GB development barriers, and provide a comprehensive reference for future academic researchers.

## 2. Methods

This paper combines academical articles and conference proceedings by keyword searching and original contents and data from official web sites of green building evaluation standards in various countries. Relevant literature reviews of green building development mainly use multiple databases like Web of Science and Scopus [12–14]. Some researchers believed Scopus is better in terms of accuracy [12], and also had a wider range of academical literature coverage [15]. They used Scopes to identify the paradigms of GB research and draw the trend of GB development. Some authors use keyword searches to collect relevant articles. Likewise, this study adopted databases and keyword searches to identify relevant articles of GB and technologies. Additionally, the original official politics of different countries, and GB rating systems all over the world and their current development status were reviewed as well. The contents and data are mainly from the official web site. Some of them are translated from the local language to English.

The method of this paper consists of six elements, of which the structure is shown in Figure 1. The first is to identify all factors that influence the development of green building in the world and divide them into two categories—the external and internal. The purpose of this division is to clearly identify the key influencing factors related to different stakeholders in the development of green buildings. The second is to study the history of GB to understand the original purpose of the concept which is designed to deal with the global energy crisis and environmental problems. It attracted considerable interest from fields as diverse as architectural engineering technologies, economics, human health, and assessment methods over time. The concept continues to develop with a range of opinions. The third is to analyze all influence factors. The external factor refers to the development status of green building, which includes policy support, economic benefits, and certification schemes. A clear roadmap is provided by analyzing these three factors for policy formulation and national standard development. The internal factor refers to fundamental characteristics of green building which include technologies implementation, building management, and occupants' behavior. The study of these factors is to offer guidance for designers, engineers, and all stakeholders to deal with GB development barriers. Finally, future trends and tendencies provided a comprehensive reference and potential directions of related studies for future academic researchers.

The structure of this paper is shown in Figure 1. Firstly, a comprehensive survey of the historical and current development of GB is summarized in Section 3. The status quo of relevant GBs policies, certification standards and projects achievement in various countries, which stand for external factor, are surveyed and summarized in Section 4. Following that, the internal factor in terms of a detailed fundamental state of GBs with specific technologies is introduced in Section 5. Subsequently, Section 6 focuses on the barriers to the adoption of GBs and strategies for overcoming these barriers. Finally, the conclusion is provided in Section 7.

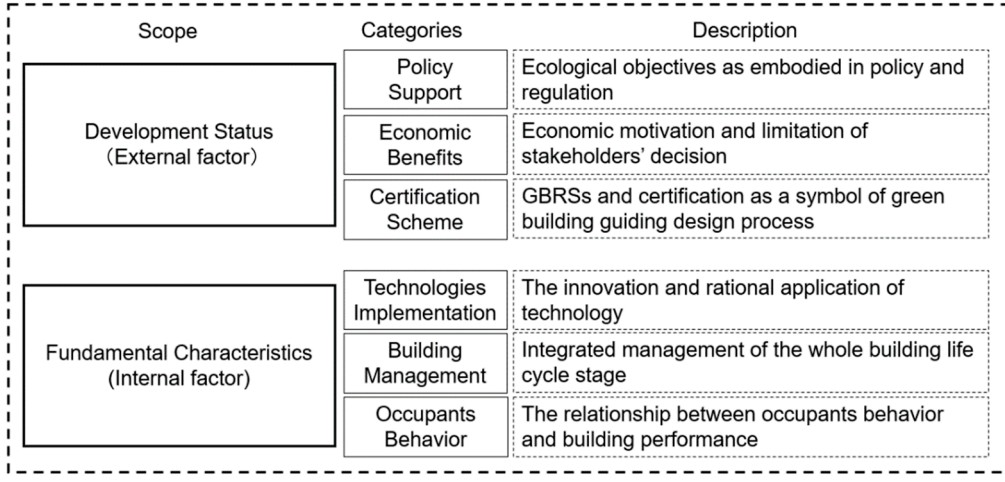

**Figure 1.** Factors influencing green building (GB) development.

## 3. Background and Definition

Green building development can be traced back to the energy crisis in the 1960s, which spurred crucial research and activities to improve energy efficiency and decrease environmental pollution [16]. Combined with the energetic environmental movement of the time, these early experiments led to the contemporary GB movement, which originated from a focus on energy-efficient and environmentally friendly building construction practices. The Earth Summit held in 1992, also known as the United Nations Conference on Environment and Development (UNCED), brought forth the Rio Declaration on Environment and Development and Agenda that stimulated the building environmental protection upsurge [17]. In 1990, the first GB rating system, the Building Research Establishment Environmental Assessment Method (BREEAM), which was developed by the Building Research Establishment (BRE) in the UK, presented a systematic method to evaluate the implementation and performance of GBs [18].

Following this point, extensive GB assessment tools were developed by government or third parties of different countries with the aim of addressing the quality of buildings [19].

Green buildings are not easily defined, as the concept continues to develop with a range of opinions. The World Green Building Council (WorldGBC) is a global network of GB councils in over 70 countries. It claims that countries and regions have various characteristics such as history, culture and traditions, distinctive climates conditions, different building types and ages, and environmental, economic, and social priorities that shape GB methods [20]. Green building is not the same across the globe [21]; its definitions represent the requirements of national and regional building industry development. WorldGBC defines green building as aiming to reduce or eliminate negative impacts on the environment during the whole building life cycle, creating positive impacts on the climate and environment [22]. The United States Environmental Protection Agency (EPA) has claimed that "green building is the practice of creating structures and using processes that are environmentally responsible and resource-efficient throughout a building's life cycle from siting to design, construction, operation, maintenance, renovation, and deconstruction [11]." A generally accepted description in the United Kingdom and European Union is that a green building contributes in some way to preserving the environment, while also considering the idea of well-being of the occupants, both in terms of use of space and quality of air. The concept is closer to that of sustainable buildings and sustainable construction. Apart from energy efficiency, it also includes aspects such as the decrease of $CO_2$ emissions, which seems to differ slightly between the EU and the U.S. [23]. The first GB certification system, BREEAM, could represent the concept of GBs in the UK that focuses both on energy efficiency and the well-being of people who live and work in the building [24]. This concept makes green and sustainable buildings interchangeable. Similarly, the GB definition in Japan also shares the meaning with sustainable building, by including energy and resources, materials, and emission of toxic substances, while also seeking to harmonize the building with local aspects and improve human life [25]. Table 1 indicates a selection of GB definitions from different organizations.

**Table 1.** Definitions of GBs.

| Country | Organization | Definition |
|---|---|---|
| USA | World Green Building Council | A GB is a building that, in its design, construction, or operation reduces or eliminates negative impacts, and can create positive impacts, on our climate and natural environment [20]. |
|  | U.S. Environmental Protection Agency (EPA) | Green building is the practice of creating structures and using processes that are environmentally responsible and resource-efficient throughout a building's life cycle, from siting to design, construction, operation, maintenance, renovation, and deconstruction [11]. |
|  | U.S. Green Building Council (USGBC) | The planning, design, construction, and operations of buildings with several central, foremost considerations: energy use, water use, indoor environmental quality, material use, and the building's effects on its site [26]. |
| UK | Building Research Establishment | The GB Certification BREEAM could represent the concept of GBs that are more sustainable environments that enhance the well-being of the people who live and work in them, help protect natural resources, and make more attractive property investments [24]. |
| Europe | European Commission Delegation | A Sustainable Building contributes in some way to preserving the environment, also increasingly extends to the idea of the well-being of the occupants, both in terms of space usage and air quality [23]. |
| Germany | German Sustainable Building Council (DGNB) | Sustainable building means using and introducing available resources consciously, minimizing energy consumption and preserving the environment [27]. |

**Table 1.** *Cont.*

| Country | Organization | Definition |
| --- | --- | --- |
| France | Haute Qualite Environment (HQE) | Certificated sustainable building endorse the overall performance of a building and that of the four areas considered by the certification scheme: energy, environment, health and comfort [28]. |
| Australia | Green Building Council Australia | Green Building incorporates principles of sustainable development, meeting the needs of the present without compromising the future [29]. |
| Japan | Architectural Institute of Japan (AIJ) | A sustainable building (green building) is one which is designed: (1) to save energy and resources, recycle materials, and minimize the emission of toxic substances throughout its life cycle; (2) to harmonize with the local climate, traditions, culture, and surrounding environment; and (3) to be able to sustain and improve the quality of human life while maintaining the capacity of the ecosystem at the local and global levels [25]. |
| China | Assessment Standard of GBs | Green building refers to a building that saves resources to the extent within the whole life cycle of the building, including saving energy, land, water, and materials while protecting the environment and reducing pollution so it provides people with a healthy, comfortable, efficient use space, and works in harmony with nature [30]. |
| Singapore | Inter-Ministerial Committee on Sustainable Development (IMCSD) | Green building is energy and water efficient, with a high quality and healthy indoor environment, integrated with green spaces and constructed from eco-friendly materials [31]. |

Some researchers wanted to demarcate the concept of GBs and sustainability in detail. However, that approach will lead to a narrow understanding of GBs that limit their development. They think although GBs have been developing, the environmental aspect is the core concept [7]. GBs are environmentally and ecologically sound in terms of land, energy, water, and materials. Sustainability is a nonstop development concept that depends on various countries' building practices [32]. It consists of four aspects: environmental, social, economic impacts, and institutional dimension [7,33]. According to different development situations, the concept of sustainability could contain every factor of human activity [34]. Whereas, focusing exclusively on the energy conservation and environmental aspect but neglecting the social, economic, and institutional factors will hinder GB development. At present, although many GB concepts have been successful and are developing in a good direction, there are still many obstacles and misunderstandings about GBs. Section 6 discusses this in more detail.

## 4. Development Status

The development status of GBs relates to external factors, including policy support, economic benefits, and certification schemes. Ecological objectives are embodied in policy and regulation. Economic benefits will influence the motivation of a stakeholders' decision. The green building certification scheme's purpose is to be a symbol, and as a green building guide for the construction process.

### 4.1. Policy Support

As noted above, GB is an integrated process of the whole building life cycle, with many components, including energy, water, materials, land, environment, human health, construction, management, and more. Any policy related to these areas can be further related to GBs. The GBs in the United States, the UK, and Japan have entered a relatively mature implementation stage. Those countries have established and improved the GB laws and regulation systems. These laws, regulations, departmental codes, and regional regulations of GBs depend on and complement each other. The perfect and comprehensive legal system provides an important guarantee and premise for the standard development of GB.

In the United States, the GB policies include mandate and incentive-based policies, which both play vital roles in GB implementation [35–37]. The government adopts zoning regulations and building benchmarks to guarantee the realization of GBs objectives. They can be classified at the federal, state, and local levels [37,38]. Policies at the federal level are mainly for buildings constructed and occupied by the government. They always focus on internal activities, with the aim to decrease the environmental footprint; examples of these are the Energy Policy Act of 2005 and The Federal Green Construction Guide for Specifiers [39]. Green policies at state levels focus on non-government buildings and require volunteer efforts by private developers [39]. However, some policies cannot adequately pursue local GB objectives. Consequently, many local governments establish their own green policies which are more detailed and likely to promote the involvement of private developers [39,40]. The incentive-based policies are grouped with various strategies, such as tax incentives, financial incentives, density bonuses, and priority permit processing to achieve an environmental agenda. In 2000, the State of New York first adopted a tax-based incentive program for GBs. Many states integrated their financial incentives to the third-party verification system, such as Oregon and Maryland. Following the Oregon statutory directive, the State Department of Energy employed Leadership in Energy and Environmental Design (LEED) as the applicable standard to help a project get a tax credit [39]. California instituted GB guideline in 2004 as the first mandatory policy. The City of Chicago proposed the Chicago Standard, which asks all new municipal construction meet LEED certification. Regulation is regarded as the most powerful policy tool for GB development [41].

There are 60 results in guidance, regulation, and business funds and grants for energy efficiency in buildings from 2008 to 2018 in the UK government website [42]. Building regulation guides the British construction industry, which sets the minimum performance standard for energy-saving performance of buildings, utilization of renewable energy, and carbon emission reduction [43]. The implementation of the building energy efficiency label is one of the effective measures to promote GBs in the UK. Additionally, the British government commissioned the British Research Establishment (BRE) to develop the Sustainable Housing Code, which is a mandated standard that guides the building industry in implementing GBs. Furthermore, since 2008, all new homes in England and new homes funded or recommended by the government and authorities in Wales, as well as all-new independent public rental housing in Northern Ireland, will be subject to a mandatory building rating process. BREEAM is widely applied in the UK, due to the fact that professional organizations and the construction industry have made a great effort to progressively make it compulsory for all new buildings and renovation projects [44]. In November 2018, the European Commission presented its strategic long-term vision to reduce greenhouse gas (GHG) emissions, showing how Europe can lead the way to climate neutrality, an economy with net-zero GHG emissions [45]. Sustainable and climate-proofed buildings are required to meet the targets to achieve a climate-neutral Europe by 2050.

Japan is a country with very limited energy and resources. Energy security is always the most significant issue in Japan, especially with the serious global warming problem. Consequently, the Japanese government has been making unremitting efforts to guide the national building energy conservation work and the promotion of GBs through laws, regulations, and policies. Japan has a wide range of relevant laws, regulations, and policies that they keep updating based on development. The policies include mandates, supports, and incentives. In 1979, Japan formulated the Energy Conservation Law, which holds up the basic principles of energy conservation. It strengthens the independent energy management of enterprises. Simultaneously, it standardized the energy-using management relationship and energy-saving behaviors among government, enterprises, and individuals, which provided the working basis for energy conservation management in Japan. The government established standards for constructors to promote the use of energy-saving measures in home construction. For building sellers and renters, it is clearly stipulated that they must provide information to consumers by energy-saving performance labeling. Moreover, the government offered financial incentives that encourage both the GBs construction and development of advanced building technologies. Green



retrofits can also earn incentives. The government leads the promotion of the GB rating system (CASBEE), which is jointly developed and promoted by industry, universities, and research institutes.

Due to the dramatic construction boom and rapid urbanization, GBs in China have significant implications [46,47]. In 2013 the Chinese government issued the Green Building Action Plan, which accelerated China's GB development and promoted the transformation of the development mode of the construction industry. One billion square meters of GBs are expected to be completed from 2015 to 2020. The percentage of certificated GBs area to new urban buildings construction area was 20% in 2015 and is expected to be 50% in 2020. Meanwhile, China emphasizes the development of GBs through a combination of mandates and incentives. Some local governments mandated that all new construction of public buildings meet the requirement for GBs. For example, Shanghai has passed local legislation to establish a mandatory promotion system, stipulating that all new buildings in the city shall comply with the GBs standards. No less than 70% of new public buildings in low-carbon development practice areas and key functional areas are constructed according to the two-star standard or above. The strictest water resource management system is implemented, controlling, and managing the total amount of water used by regions and enterprises. Shanghai vigorously promotes water-saving demonstration activities in water-saving parks, campuses, communities, enterprises, and government agencies. By 2020, the water consumption of 10 million yuan of GDP and 10 million yuan of industrial added value in the city will decrease by about 23% and 20%, respectively, compared with 2015 [48].

Finland's industries have set ambitious targets for 2030 that will triple the market share of wood construction, double the value added to the woodworking industries, and decrease the environmental impact by 30% [49]. In Australia, to fulfill the commitment to reduce up to 28% of GHG emissions by 2030 [50], many green-building rating tools have been developed. In India, some government agencies have provided discounts on premium charges. The Ministry of New and Renewable Energy (MNRE) mandates that all government buildings should be at minimum Green Rating for Integrated Habitat Assessment (GRIHA) three stars certified [51]. The Malaysian government has facilitated communication between the private sector and non-profit organizations [52]. Certified GBs can apply for tax and stamp duty exemptions [53]. Eligible GBs in Singapore can get up to a 2% gross floor area (GFA) bonus [54]. In Indonesia, the Quezon City Government passed its GB Ordinance No. SP-1917 (QCGBO) in 2009. All the new buildings and retrofit structures in Quezon City must comply with the Implementing Rules and Regulations (IRR) of the GB ordinance [55]. Green buildings in Vietnam are still in their infancy, and facing numerous challenges. Similar to the Singapore Building and Construction Agency, the Vietnam government is developing its own agency to promote GB projects and improve the efficiency of the decision-making framework for GB development. Hanoi and Ho Chi Minh City will be the first pilots before the decision-making model is applied to the whole country [56]. Since 2009 when the Vietnam Green Building Council (VGBC) was endorsed to develop LOTUS, a set of market-based green building rating systems specifically for the Vietnamese built environment, there has been a continuous increase in awareness of green building benefits among policymakers, investors, and industry professionals. The National Green Growth Strategy, which was issued by the Prime Minister of Vietnam, indicated that the government "require investors to implement green measures when they build new commercial buildings or retrofit old buildings, and will have incentives for manufacturers who make products for green buildings" [57].

The process of promoting GB implementation is slightly different between Western and Eastern countries. In Eastern counties, such as Japan and China, the government organizes the formulation of relevant standards and implements them gradually; even adopting mandatory measures to conduct strict management from the planning and design stage of buildings. Western countries such as the United States differ from this model, adopting federal, state, and local level zoning regulations and employing building standards developed by non-governmental organizations. As the first country to implement green building certification, the UK has achieved a relatively advanced level of green building development. Ethical consideration has also played an important role in the development of green buildings. In addition to the relevant policy and economic support, what is more important is

that the UK's professional organizations and industrial construction will take sustainable development as their social responsibility. They all see it as their responsibility to develop green buildings, not just for financial support or certification labels.

*4.2. Economic Benefits*

Some recent research has focused on the economics of GBs, which is one of the most important factors influencing stakeholders' GB implementation decisions. Ofek et al. (2018) explored factors influencing the investment decisions of three GB interest groups—consumers, architects, and building developers in Israel. They found that potential energy and maintenance savings and increases in real estate values are the main forces driving consumers' decisions [58]. Maintenance savings are one of the vital factors positively related to GB premium size [59]. By contrast, energy price increases and striving for innovation are the main factors influencing developers' decisions [58].

There is a common idea that high technology means high price and that GBs equal high-cost buildings. Some researchers argue that certified GBs cannot save money or even energy. On the contrary, others believe that GBs can contribute significantly to energy and money savings, and provide environmentally friendly construction.

Green building projects added extra costs of 1% to 10%, based on Lockwood's research. This is because the green premium includes efficient mechanical systems which are quite expensive and complex extended designing process [60]. Dwaikat and Ali (2018) used the life cycle cost (LCC) method and found that the future cost associated with operation and maintenance is 3.6 times higher than the initial cost of GBs [61]. Davis Langdon (2007) indicates that the initial construction costs of a five-star Green Star building are likely to be 3% to 5% higher than conventional buildings, and 8% to 10% or a six-star Green Star project [62]. Ross et al. (2007) developed a financial model that illustrated that LEED-certified projects cost 10% more because of the large cost of labor and materials, which accounts for the largest proportion of GB costs [63].

On the other hand, from a maintenance perspective, some researchers suggest that GBs perform better than conventional buildings in terms of energy efficiency and water efficiency, which improves cost efficiency [64]. The Indian researcher Vyas (2015) outlined the potential benefit of Indian government GBs. The average increase in the initial cost is 3.1% for three-star certified GBs and 9.37% for five-star GBs. The discounted payback period for GBs, which considers the time value of money, is 2.04 to 7.56 years for three-star certified projects and 2.37 to 9.14 years for five-star ones. However, Vyas believes that savings from a GBs can cover the incremental cost in GBs [51]. Zhao (2018) investigated the time effects of GB policy on energy performance in low-income house units. Due to reduced energy usage in GBs, financial savings came to 648 dollars per year [65].

*4.3. Certification Schemes*

4.3.1. GBs Rating Systems (GBRSs)

Since the first GB assessment BREEAM issued in 1990, the development of GB aligns with the development of the green building rating system (GBRS). Over forty GBRSs have been developed by governments or third parties with the aim of promoting sustainable buildings [19,66,67]. Using the keyword 'green buildings', 'green buildings rating system', and 'green buildings standard' on the Internet, and related research papers, there are 49 rating systems summarized specifically for GB design and certification in various countries (Figure 2 and Appendix A). Approximately four-fifths of the systems are used in their own countries. A GBRS defines the attributes of GBs, provides tools to assess the environmental effects of buildings, and identifies specific interventions intended to promote the green building market [68]. Countries develop GBRSs based on the principle of adapting to local conditions and constantly update them in real-time to meet GB development needs. In addition, throughout the GB development process, GBRS institutions have played a vital role in promoting GB

development. They established a long-term, scientific GB market mechanism through open and fair GB evaluation and certification work.

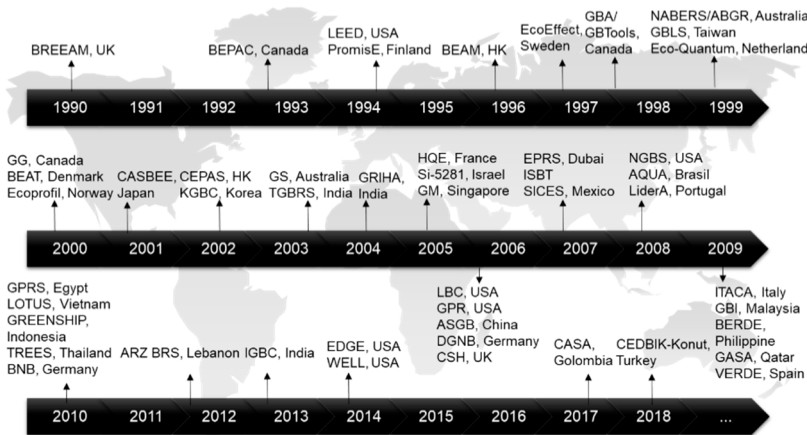

**Figure 2.** Timeline of green building rating system (GBRS) development [7,39,55,69–85].

Over the past 20 years, extensive research has focused on GBRS conditions and development. Todd analyzed the global trends in LEED certification, including LEED for New Construction Rating System (LEED-NC) and LEED for Existing Buildings: Operations and Maintenance (LEED-EBOM) and individual LEED credits achievement [68]. Ponterosso et al. compared the physically monitored environment of a BREEAM "Excellent" certification office with occupancy comfort and building management system metrics [86]. The concept and framework of Comprehensive Assessment System for Built Environment Efficiency (CASBEE)-City were introduced by Murakami [87]. While other researchers compared selected GBRSs to investigate the different indicators or their capability in promoting GB development, Li et al. (2017) proposed a four-level assessment method comparison that features: (1) general comparisons; (2) category comparisons; (3) criterion comparisons; and (4) indicator comparisons, which are based on 57 articles from three academic databases [88]. Doan (2017) compared four GB rating systems: LEED, BREEAM, CASBEE, and Green Star NZ. Indoor environmental quality, energy, and materials are core common elements of content for the four rating systems [7]. Doan indicated that 408 papers related to BREEAM, LEED, or CASBEE were published in various professional journals since 1998. The number of GB rating papers increased dramatically from 1998 to 2006. Compared to the significantly higher number of papers discussing LEED and BREEAM, the number of research papers about CASBEE and GREEN Star NZ is limited [7].

Many evaluation criteria have developed a series of sub-evaluation systems tailored to different scales, construction phase, or building type. For example, LEED includes LEED Building Design and Construction (BD + C), LEED Interior Design and Construction (ID + C), LEED Building Operations and Maintenance (O + M), LEED Neighbourhood Development (ND), LEED Homes, and more. CASBEE consists of construction (housing and buildings), urban (town development), and city management. According to the Construction phase, BREEAM is divided into New Construction (NC), BREEAM in-use, BREEAM Refurbishment and Fit-Out (RFO). China's GBRS family includes Green Commercial Building, Green Industrial Building, Green Hospital, Green Museum, and more, classifying subcategories based on building types. These standards will be more targeted to give the appropriate GB construction strategy for a select building type.

### 4.3.2. Accredited Professionals (AP)

For better GBRS implementation, many professionals who conduct auditing for achieving GBRS credits were certified. Sometimes they also can help to implement the international application of the GBRS to which the professionals belong. They work closely with the design team and the developers during the entire building construction process. The workflow is shown in Figure 3.

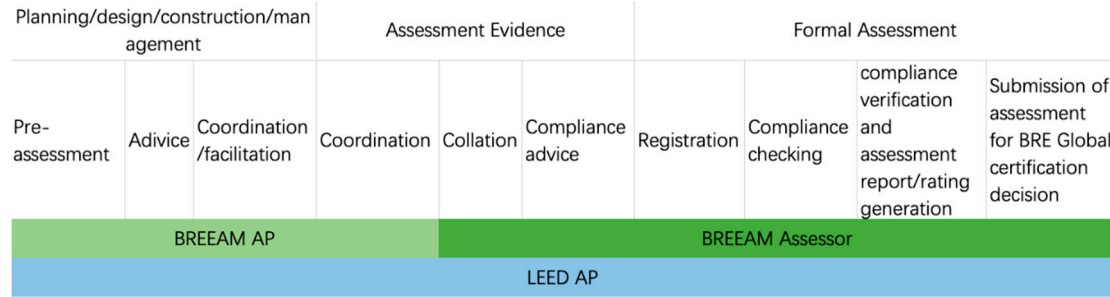

**Figure 3.** The certified accredited professionals workflow [89].

Of the 49 rating systems in the world, 18 GBRSs have developed an accredited professionals' certification (Table 2). The other standard systems do not specify the qualification requirements for evaluators. The admission requirements of GBRS professionals are similar. First, BREEAM, German Sustainable Building Council (DGNB, Stuttgart, Germany), High-Quality Environmental standard (HQE, Paris, France), Green Mark (GM, Singapore, Singapore), Built Environmental Assessment Method (BEAM, Hong Kong, China), and GRIHA ask that those applying hold a university degree or an equivalent qualification from the construction field with professional working experience. CASBEE even requires that the applicant for AP of new building design and construction hold a first-class architect license [90]. The other GBRSs in Table 2 strongly recommend degree-level education and working experience but it is not compulsory. Almost all the eligible applicants must participate in relevant training courses, in person or online, or take part in a workshop initially. After completing all the aspects of the training courses, they will learn the role of the AP and what a project or development needs to do to meet GBRS targets and sustainability goals. Following the training, the students must pass an examination so they can attain the certification of accredited professionals. LEED does not require participation in the training course, but requires that applicants pass the LEED Green Associate (LEED GA) test first, and then take and pass the LEED AP test, to be granted the LEED AP certificate and use of the industry logo. GREENSHIP in Indonesia is similar to LEED GA and AP [91]. There are no prerequisites or eligibility requirements for the LEED GA examination. In the LEED and Green Star evaluation process, projects involving LEED AP or Green Star AP will achieve an additional credit.

**Table 2.** Certification requirement for professionals of GBRSs [89–106].

| Countries | Standard | Professionals | Education | Working Experience | Training Course | Examination | Extra Credits |
|---|---|---|---|---|---|---|---|
| America | LEED | LEED GA LEED AP | ○ | ○ | ○ | ● | ● |
| | GPR | Certified GB Professional (CGBP) | ○ | ○ | ● | ● | × |
| | EDGE | EDGE Expert | ○ | ○ | ● | ● | × |
| | WELL | WELL AP | ○ | ○ | ○ | ● | × |
| United Kingdom | BREEAM | BREEAM Assessor BREEAM AP | ● | ● | ● | ● | × |
| Germany | DGNB | DGNB Registered Professional DGNB Auditors DGNB Consultant | ● | ● | ● | ● | × |
| France | HQE | HQE Référents | ● | ● | ● | ● | × |
| Australia | NABERS/ABGR | Accredited Assessor | ○ | ○ | ● | × | × |
| | GS | Accredited Professional | ○ | ○ | ● | ● | ● |

**Table 2.** *Cont.*

| Countries | Standard | Professionals | Education | Working Experience | Training Course | Examination | Extra Credits |
|---|---|---|---|---|---|---|---|
| NGO | LBC | Living Future Accredited | ○ | ○ | ● | × | × |
| Japan | CASBEE | CASBEE Accredited Professional (AP) | ● | ○ | ● | ● | × |
| Singapore | GM | Green Mark Manager (GMM) Green Mark Professional | ● | ● | ● | ● | × |
| Hong Kong | BEAM | BEAM Professionals (BEAM Pro) | ● | ● | ● | ● | × |
| Philippine | BERDE | Certified BERDE Professionals | ○ | ○ | ● | ● | × |
| Malaysia | GBI | Accredited GBI Certifier | ○ | ○ | ● | ● | × |
| India | GRIHA | GRIHA Certified Professional | ● | ● | ● | ● | × |
| Abu Dhabi | EPRS | Pearl Qualified Professional | ○ | ○ | ● | ● | × |
| Indonesia | GREENSHIP | GREENSHIP GA GREENSHIP AP | ○ | ○ | ● | ● | × |

Note: ●: mandatory; ○: strongly recommend; ×: None.

### 4.3.3. Project Achievements

Since BREEAM was promulgated in 1990, it has been carried out in 77 countries for nearly 30 years, with a total of 565,790 certification programs accumulated, ranking the first in the world, accounting for 80% of the total certificated green building projects in the world (Figure 4). LEED, which was enacted in 1998, has the widest reach, reaching 167 countries [107]. WELL followed BREEAM as the third widest used in 58 countries [108]. Excellence in Design for Greater Efficiencies (EDGE, Washington, America), DGNB, and Living Building Challenge (LBC, Seattle, America) are used in more than twenty countries [70,109,110]. HQE, the Green Building Assessment (GBA, Ottawa, Canada), and GM are used in 17, 16, and 15 countries, respectively. Assessment Standard of GBs (ESGB, Beijing, China), and Green Globes (GG, Toronto, Canada) are tentatively being applied in one country outside their own countries (Figure 5). According to the SmartMarket report "Global GBs Trends 2018" jointly released by Dodge Data & Analytics and U.S. Green Building Council (USGBC), the global GB market is on the rise. Of the respondents in the study, 47% believe that more than 60% of their construction projects will be certified under a recognized green building system by 2021. Nineteen of these countries are expected to see strong growth over the next three years. The report surveyed more than 2000 building experts in 86 countries, including architects, contractors, consultants, developers, engineering companies, and investors. Nearly half of those surveyed said they would focus on GB projects over the next three years. Market demand and health factors are key drivers of the building sector's transition to sustainable development, with the future growth of new commercial buildings, institutions, and high-end residential buildings particularly promising. Two-thirds of respondents also said LEED certification makes buildings perform better, while more than half said LEED provides credibility for GBs. Almost two-thirds of those surveyed predicted that GBs would save 6% on operating costs over the next year, with 80% saying the trend would continue over the next five years. With the popularity of operating costs and health benefits, the value of GBs will continue to increase [20].

The importance of technology in GBs has always been underestimated, particularly in measuring energy performance and its impact on households. In 2016, Green Business Certification Inc. (GBCI) (the global LEED project certification body and a green enterprise certification company) created the Arc certification platform to manage and compare building data through five measurement criteria: energy, water, waste, transportation, and human experience. Tracking performance is the key to future

GB certification. Both Arc and LEED v4.1 are designed to provide a quick and easy way to create a healthy living environment to ensure that all GBs perform well from the start of construction to completion and beyond. Arc has now certified 1.5 billion square meters in 80 countries worldwide. The LEED v4.1 rating system introduced in 2019 also provides a new way to improve GB performance. At present, there are 94,000 LEED-certified commercial projects around the world, with an average of 2.2 million square meters of LEED-certified buildings every day.

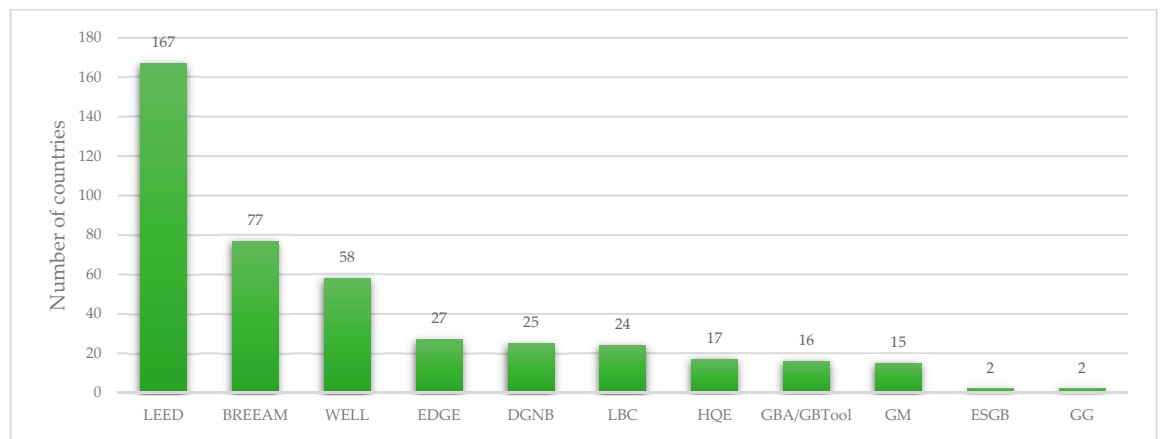

**Figure 4.** The number of countries in which each standard is applied (by 2018) [70,109,111–119].

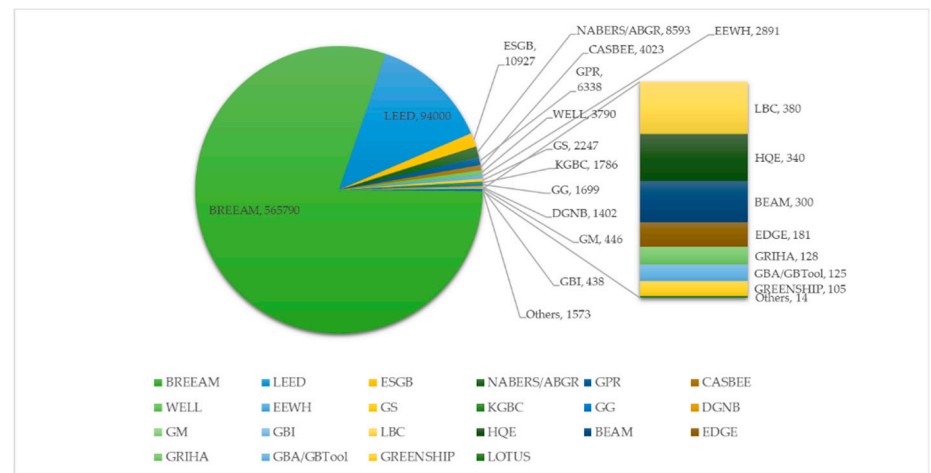

**Figure 5.** Certified GBRS projects [71,73,108,109,111–113,115–128].

## 5. Fundamental Characteristics

The fundamental characteristics of GBs stand for internal factors, including GB technologies (GBTs) implementation, building management, and occupant behavior. The term *GBTs* refers to technologies integrated into building design and construction to make the building sustainable [129,130]. Managerial aspects of green buildings refer to integrated management of the whole building life cycle stage [131]. The third is the relationship between occupant behavior and building performance.

### 5.1. Technologies Application

Adopting GB technologies can offer a range of significant environmental benefits, such as saving land and materials, increasing the efficiency of water and energy usage, and improving indoor environmental quality [13,85,130,132]. There are extensive studies on various aspects of GBTs in different contexts. Yin and Li developed a stochastic differential game that transfers GB technologies from academic research institutes to building enterprises in the building enterprises-academic research

institutes collaborative innovation (BACI) system, which will promote GB technology transfer and rapid development of urban GBs [133]. Comparing the 49 green building evaluation standards, six categories of land use, energy conservation, water efficiency, material utilization, indoor environment quality improvement, and construction management are common significant technologies implemented in green building construction.

General land use measurements mainly solve three issues, how to use properly, how to save efficiently, and how to improve effectively. Firstly, in the perspective of architects and landscapers, outdoor open space and green space for occupants' activities, enough parking space for the increasing usage of cars, and outdoor microclimate design strategies to support natural ventilation and natural lighting for the buildings' indoor environment are vital for the 'use properly' issue. Sharing public facilities is also a method to use land properly. Secondly, with the increasing requirement of space for occupied because of increasing population and rapid urbanization, especially in China, for instance, one of the methods to save the land is setting a high plot ratio objective, which means high-rise building is increased. Limited land area is another reason that requires saving of land. For example, in Tokyo, people try to give multiple functions to limited building space, and design alterable space for various requirements. Thirdly, the ecological protection of construction sites is significantly important as well. Most of the GBRs have claimed that they try to keep the original ecological system of construction sites, avoiding construction in wetland, habitat, etc.

Building energy conservation measures include three aspects of buildings: envelope, air conditioning, and lighting. Generally, architects, mechanical engineers, and electrical engineers, respectively, are responsible for these three parts. Aktacir, Büyükalaca, and Yilmaz (2010) evaluated the influence of thermal insulation on the building cooling load in Adana, which showed that both the initial and the operating costs of the air-conditioning system were decreased considerably for three evaluated insulation thicknesses [134]. Air conditioning contributes to maintaining thermal comfort, which accounts for a major share of energy consumption. Chua et al. reviewed technologies and strategies for achieving better energy-efficient air conditioning, which can be divided into three aspects: novel cooling devices, innovative systems, and operational management and control [135]. The use of renewable energy technologies has been pivotal for achieving GB goals and certification [131,136]. According to Chan's research, the photovoltaic system not only generates electricity, but also reduces heat gain transmitted into the indoor environment through the building envelope by 13.59% to 38.78% in subtropical Hong Kong [137]. Passive design is believed to have big energy-saving potential. Oropeza-Perez and Østergaard (2014) investigated the energy-saving potential of natural ventilation and indicate that average savings can correspond to 54.4% of the electric cooling demand for 2008 in Mexico [138]. A simulated model to evaluate life cycle GHG emissions of office building envelopes has been developed in Australia, and that model can be used to evaluate the relationship between building energy consumption and GHG emissions to achieve the "greenest" outcomes [139].

Similar to energy aspects, water conservation is also vital in GB design due to the limitation of potable water by only 3% of the total earth's surface water [140]. GB are sustainable buildings demanding the water conservation and preventing pollution and recycle treated water ensuring potable water use. It can be divided into outdoor and indoor water use. Architects, landscapers, and engineers engaged in water supply and drainage engineering are responsible for this work in building construction process. Water efficiency refers to reducing the usage of water as well as minimizing wastewater. All the fixtures such as taps, toilets, showerheads, urinals, etc. should be efficient and be checked periodically for leakage and for good operating conditions [141]. Rainwater harvesting is a cheap and simple technology that can save a lot of water if rain can be collected and treated as potable water. The basic system consists of the collection, distribution, and storage stages. A quantity of non-potable water for water closets, car-washing, and garden watering can come from collected and treated greywater passing through sand filters, or by electrocoagulation techniques. Some other biological and chemical treatments can be utilized as well. Rainwater management is to keep the rainwater stay in the construction site rather than allowing it to run off, which not only

benefit for rainwater harvesting but protecting the natural site hydrology conditions. Low-impact development (LID) and green infrastructure (GI) are widely used in rainwater management strategies and techniques [142].

Building materials affect the environment and the human body in all stages of their life cycle production, based on contamination and function [143,144]. Firstly, the evolution process of material selection pays more attention to green and sustainable performance criteria, more than just quality, performance, aesthetics, and cost. Initiatives that have been taken and are being taken from the academic and scientific field to mitigate the effects of climate change associated with the activity of the construction sector. For example, García et al. (2019) have developed more sustainable construction systems, through the replacement of conventional concrete or steel construction elements with timber elements [145]. There are simple and rapid sustainability assessment models specific to timber structures and buildings, whose objective is to design and project timber buildings in the most sustainable way possible, with the ultimate goal of reducing the impact that the construction sector activity has in the environment [146]. Secondly, storage and collection of recyclables material is another important consideration in GB construction. Furthermore, building product disclosure and optimization is a major content in the credit category of Material and Resource. However, it is a big challenge for many other countries because of the complex supply chain management process of building products.

Indoor environment quality is an important variable for GB performance, and its improvement contributes dramatically to GBs and a sustainable environment [147]. Most researchers believe the certified GBs perform better than conventional buildings in terms of IEQ and energy use [148–150]. There are four main variables highlighted in GBRSs to improve IEQ: thermal quality, acoustic quality, visual quality, and indoor air quality (IAQ) [151–153]. Lin et al. found the satisfaction of users in certified GBs is higher than conventional buildings in terms of thermal comfort and IAQ [154]. The view to the outside, aesthetic appearance, less disturbance from heating, ventilation, and air-conditioning noise, and other factors have better outcomes as well [155].

## 5.2. Construction

Construction waste minimization (CWM) is a vital aspect of GBs construction. Lu et al. (2018) has ascertained the effects of GBs on CMW and identified the causes leading to the ascertained effects using quantitative "big data" from government agencies [156]. Building information modeling (BIM) is becoming the central way to coordinate project design and construction activities. EI-Diraby, Krijnen, and Papagelis (2017) built an online system that enables a data-driven approach to building planning, construction, and maintenance, which allows all the stakeholders to comment and share views [157]. Lu et al. (2017) provided a "green BIM triangle" classification to establish an up-to-date synthesis on the nexus between BIM and GBs, indicating that the relationship needs to be understood from three dimensions: project stage, green attributes, and BIM attributes [158].

## 5.3. Building Management

The managerial aspects of GBs should be integrated into the whole building life cycle, including planning, design, construction, operation, and demolition. Initially, during the planning phase of the project, the research and analysis related to energy and water use should be completed. Meanwhile, it should conduct effective and rational discussions about possible integrated design opportunities. Additionally, the project owner can be invited into the main project team workshop to determine the budget, schedule, functional planning requirements, scope, quality, performance, and desired project objectives of the occupants. In the design and construction phase, project team members look for synergies between systems and components. This combination of advantages can help the building achieve a high level of performance, comfort, and environmental benefits [142]. Constantly monitoring and studying building performance in the operation phase is just as important as it is in the design and construction phase [159]. Feedback mechanisms determine whether or not performance goals are

being achieved. To achieve those goals, it is critical to provide operational performance information to building operations staff so they can take corrective action when targets are not met. Implementation of an environmental management system (EMS) in the operation phase contributes to a 90% energy saving and 70% water saving, reduces 63% of waste, and lowers accident rates by 20% and 80% of quality complaints from occupants [160]. Management in POE to find out causes of performance gap between the design prediction and actual consumption. The actual performances always worse than the predicted. For example, glass box buildings are notoriously uncomfortable regardless of their very large, sophisticated, expensive, and maintenance-intensive system. Architectural designers do not always recognize the high probability of thermal discomfort in glass buildings in a hot climate and it often results in higher energy consumption and running costs for the business or to the owner.

*5.4. Occupant Behavior*

Along with GB development and building energy and environmental improvements, people are paying increasing attention to the relationship between people and buildings. The concepts and disciplines of a healthy building, post-occupancy evaluation (POE) [161], human factors (ergonomics), and architectural psychology have gradually become the focus of research. Organizational commercial buildings generally adopt centralized control of the electrical equipment. Occupant behavior has little influence on building performance. However, for individual residential or office buildings, occupant behavior has a very big impact on architectural performance. Barbosa and Azar give a concept human-in-the-loop approach, which means occupants' comfort and well-being are essential metrics in evaluating building performance, not only energy conservation. Green buildings are believed to be associated with high workplace satisfaction and working productively and creatively [162]. Ries et al. found a 25% growth of productivity when occupants moved from conventional building to a GB [163]. Furthermore, occupants assigned higher acceptance and satisfaction to an indoor environment in a certified GBs compared to conventional buildings [164]. In the operation phase, building performance mainly depends on the occupants, who will help achieve the initial ecological objectives by correctly using devices through a better understanding of GBs.

## 6. Discussion

*6.1. Barriers and Challenges*

6.1.1. Challenges in Various Countries

There are three main problems facing GB development in the United States. First, although the government has relatively complete policy support, and the rating systems are widely used in the world, the industry and the public remain doubtful. Some people believe that GBs have not achieved what it promised. These promises include realizing energy conservation. LEED-certified commercial buildings do not display significant primary energy savings over comparable non-LEED buildings on average, not even showing a reduction in GHG emission associated with building operation [165]. Second, the enthusiasm of architects and designers are not high because most of the policy and economical support is for developers. Architects, as the initial participants and designers of architectural construction, directly determine the basic characteristics and performance of the building. Designers' personal interests, such as capital benefit, enthusiasm for GB application, or social responsibility as a promoter of GB for public is vital for GB implementation. Some architects only design GBs according to the standards but lack understanding of the connotation of GBs and the analysis and application of appropriate technologies. Third, there is a substantial problem in how to persuade the users to buy a GB with extra expenses due to certification fees and other additional active technologies expenses.

In the UK, the situation is better than in America. As the first country to use the Green Building Rating System, the UK has formed awareness in ethics for the public to build sustainably and environmentally-friendly. However, poor GB design projects still exist due to unreasonable design,

which causes higher energy consumption than non-certified buildings. Improving architects and designers understanding of the connotation of GBs and the analysis ability on the application of appropriate technologies is significantly important. Europe has presented many concepts related to GB, such as nearly zero energy building (NZEB), and carbon-neutral building (CNB), to address climate change. Great challenges will be accompanied by the realization of the goals. For example, disconnection between developing innovative technologies for GBs and the lack of utilization, lack of understanding of what GBs, NZEB, or CNB means in legislation for the actual building process, and energy targets for green retrofitting of existing building, especially of culture and historically significant buildings, etc., are major challenges Europe is facing.

Japan's GB projects realized many achievements and essentially met its original targets. However, the requirement that CASBEE AP need to hold the first-class architect license will limit the popularity of the GB concept to stakeholders. Moreover, how to interact with end-users and persuade them to recognize the value and real benefits of GBs is significant in the continued development of GB because end-users have a limited understanding of high GB technologies or new equipment to use properly.

In China, relative to the constant introduction of various laws, regulations, standards, and norms, the implementation of incentive policies lags. The concentration of GBs is not spread evenly across the different provinces because of the geographic variables, economy-related variables, and public policies associated with GBs [36]. China has imposed extensive mandatory policies on the promotion of GBs technologies recently, but some of them have not yet reached mature levels, such as prefabricated buildings, which are now heavily promoted to save materials. The public still has questions about the technology. Mandatory widespread adoption could pose potential problems. In addition to policy and economic support, it is more important to foster a sense of responsibility for sustainable development. It is the responsibility of every stakeholder to develop green buildings, not just to meet policy requirements, obtain financial support, or obtain a certification label.

### 6.1.2. Barriers of GB Development

Limitation of standards is one of the serious barriers in the external factor of GB development. Such limitations can be divided into three categories: evaluation objects restriction, inapplicability of evaluation methods, and limited professionalism of users. Although lots of GBRSs have developed sub-evaluations for different phases, scales, and types, the standard development cannot keep pace with construction development. The corresponding evaluation criteria cannot be found for many buildings. For instance, the Evaluation Standard for Green Industrial Building (GB/T50878-2013) (ESGIB) was launched in 2014 in China for assessing all industrial building types, such as heavy industry, light industry, and so on. However, modern logistics, science and technology research, e-commerce, etc. also belong to the industrial building scale. In the functional operation of these kinds of industrial buildings, no specific production process is given. However, the green industrial building standard identifies many indicators related to parameters of the production process. These indicators are not suitable for the industrial building mentioned above. Comparing with the similar functions of industrial buildings is an optional method for evaluating the sustainable level of the building, but the lack of data and the poor comparability of the chosen industries lead to an unreliable evaluation result. It is critical to develop a standard system as soon as possible that suits the different building types, including general plant and scientific research and development buildings, so GB technology promotion and evaluation on industrial construction can be standardized.

Table 2 illustrates that 16 GBRSs have their own certified AP who can advise on the construction process. These certified experts must undergo rigorous screening, training, and testing before they can be certified to participate in the program. However, the remaining 31 GBRSs have no relevant official certification process, which makes it difficult to guarantee the professional degree of GB engineers or consultants, resulting in the inability of the project to achieve sustainable success with high efficiency. On the other hand, CASBEE has the most rigorous vetting of certification experts. This effectively guarantees the green technology quality of the project but limits the way other engineers

want to participate. It will also hinder the promotion and popularization of standards, and even limits overseas promotion.

As for the third part of the external factor of GB development, economic obstacles are also significant. Transaction costs are claimed to affect the effectiveness of GBs policy significantly [166]. Marker et al. suggested that the additional costs of GB certification consultants and paperwork are the main barriers of GB development [167]. Sometimes designers and developers are unwilling to use new technologies because they use standard accounting procedures that are unable to recognize the financial advantages.

The dissemination of GBs and adaptation of GB technologies are being hindered because of some barriers, such as greater complexity, limited understanding of sustainability, and high cost [168]. Moreover, some problems have already been revealed in the GB market. Newsham found that LEED-certified buildings consume 18% to 39% less energy per floor area than their conventional counterparts on average, which is based on the comparison of 100 LEED commercial and institutional buildings to the energy use of the general American commercial buildings. Nevertheless, 28% to 35% of LEED-certified buildings are using more energy than their conventional counterparts [66]. Of the LEED-certified buildings, 25% cannot save as much energy as predicted in the design process [169]. USGBC has pointed out that the construction method of GBs is not mature enough, and the use of new GB technologies may cause potential risk. The building performance gap between design prediction and actual consumption is also required to be considered carefully. The building industry should take up these new challenges facing risk management [170].

Limitation of knowledge refers to lack of understanding about the concept of GBs used by those who can incorporate GB concepts into a building life cycle, including owners, architects, architectural engineers, construction managers, building operators, occupants, and other stakeholders. The significance of knowledge centers around three main aspects: The advantages of GBs, knowledge of existing green technologies, and cognition of how to use GBs technologies appropriately and efficiently.

First, the advantages of GBs is basic knowledge stakeholders need, otherwise, they will have no incentive to implement GBs [171–173]. Liu et al. believe elements like subjective knowledge, social trust in the organizations responsible, perceived usefulness, and the attitude of users towards green-certified buildings are among the vital psychological determinants of intention to adopt green-certified building [174]. Darko and Chan evaluated GBT adoption in developing countries and concluded that publicity through media and educational and training programs for developers, constructors, and policymakers are the top two strategies to promote GBT adoption [175]. Second, in terms of knowledge of existing green technologies, sometimes people recognize the necessity to implement GBs but lack the knowledge of which technologies are available to do that. Tsantopoulos et al.(2018) reported on the public perceptions and attitudes toward green roofs, vertical trellises, or gardens, and showed that most citizens are willing to improve aesthetics with no awareness of the environmental benefits [176]. Hobman and Frederiks (2014) conducted a large national survey with over 900 Australian energy consumers who had not to subscribed to the National GreenPower Programme and concluded that one of the main reasons was limited knowledge, awareness, and availability of the green electricity program [177]. Additionally, those who might finance the construction may fail to recognize the benefits of integration, or may mistakenly assume that existing building methods are already effective and therefore do not seem to require new technology. Third, the cognition of how to use GB technologies appropriately and efficiently is lacking. Incorrect use of technology not only precludes positive results, it also may bring a negative impact and crisis. There are several technologies implemented in GBs construction by mistake. For example, some scholars questioned whether external insulation is required in temperate and subtropical regions. There is a temperature difference between the two sides of the building walls, so heat preservation materials should be added to prevent the temperature difference from causing heat transfer to save energy. However, in a warm region, where there may only be a small temperature difference between the two sides of the building, insulation will be required less, or no insulation may be needed at all. The outdoor temperature in a warm region may often be in a range

between 18 °C~25 °C—a comfort zone. However, as the sun shines through the window, the house becomes very hot, and the lower U-value of building envelop is, the less heat will be able to escape (if the house is not well-ventilated naturally). Instead, the air conditioner needs to be turned on to cool the house, which will lead to extra energy consumption.

*6.2. Future Trends and Tendencies*

To realize the scale-up and implementation of GBs, a mountain of further effort is still necessary. According to the above review and analysis, GB development can be improved in two respects. First, from the policy and incentive side. It still requires clear and multiple policy support for the stakeholders and broad range of building types. In a word, GBs require not only environmental innovation but also institutional innovation. Second, from the economic side, cost-benefits are the most effective and direct drivers for successful GB implementation. In addition to cost savings from improved energy efficiency, the potential value added to the property should be investigated in future research. Additional costs of GB certification consultants and paperwork should receive more government support. Third, the evaluation content and application mode of GB evaluation standards need to be more rigorous and standardized. International standards should take into account the local climate and culture. The project should not adopt inappropriate technology or adopt high and new technology without considering the economic impacts and should not blindly pursue multiple certifications. Fourth, social responsibility or ethic consideration of individual and public need to be improved urgently which can fundamentally promote the development of GB.

The internal factors consist of the technology, management, and occupants. First, the technology field related to GBs is quite broad, encompassing land, energy, water resources, materials, building structure, indoor environment to construction technology, and more. Every aspect of technology development is crucial to GB development. This requires the joint efforts and cooperation of all relevant technical personnel and researchers, as well as constantly upgraded related technologies, so as to achieve the maximum benefit of technical solutions and meet the evolution of the end users' motivation and the surrounding environment. The well-developed GB technology is not only the study of a single technology but also the ability to integrate multiple technologies and enable various stakeholders to continually participate in the process of GB construction. Second, based on the implementation of multiple technologies, an integrated management methodology is necessary to handle all aspects of GBs. Currently, this role is played by GB consultants, most of whom are certified professionals. It is expected that all stakeholders can attain basic knowledge that enables them to improve the efficiency and flexibility of management systems. Third, from the occupants' perspective, enhancing their feedback is essential, because they directly impact the successful implementation of GBs. Therefore, knowledge of GBs is extremely important not only for engineers but also for occupants. In the operation phase, successful building performance mainly depends on occupants who contribute to achieving the initial ecological objective by correctly using devices because they have a better understanding of GBs. In addition, it is critical to seriously study occupants' behavior, to help human-oriented design and realize a healthier building environment. Providing training and education in using GBTs, and to develop a better awareness of local environmental issues is expected in the future.

## 7. Conclusions

This paper reports on a comprehensive survey of the historical and current development of GB worldwide. The concept of GB evolves as a holistic approach to deal with various problems caused by the construction industry. Green building is subject to continuous development of new technologies, integrated management of building operation, consistent standards of certification systems, and proper adjustment of policies, all of which have a significant impact on GB development. The method applied in this paper was to group the impact factor into two aspects: (1) external factors, including policy support, economic benefits, and certification schemes of GBs; and (2) the internal factors, associated with the development and application of GB technology, the level of building management, and how

users interact with the GB technology. Based on the external and internal factors, this paper analyzes GB development barriers and challenges.

The development status of GBs in the United States, Europe, the United Kingdom, Japan, China, and some other countries are presented in this paper. The United States, the United Kingdom, Europe, and some Western countries have already entered into a mature period. The focus of their recent work is on the application of intelligent GB technologies that ensures smart buildings or 'healthy' buildings which proposed by the International WELL Building Institute (IWBI) [178], and consequently addresses the economic and social challenges caused by unmatched technologies and limited knowledge. Japan has a wide range of relevant laws, regulations, and policies, but keeps updating them based on development. This paper has found that GBs in China have significant implications; as a national strategy, the development of GB is leading the construction field on the road of sustainable development. However, in China, there is a regional imbalance of GB development because the concentration of GBs and economic strength varies across its different provinces. In the process of promoting the implementation of GBs, Eastern counties, such as Japan and China, have mainly developed government programs. In contrast, Western countries such as the United States have adopted federal-, state-, and local-level zoning regulations and employ building standards developed by non-government organizations. Although each country has made many achievements in the development of GB, this paper also reveals that a common problem is the lack of a systematic social education scheme that can provide a clear understanding about the concept of GB to those who can incorporate it into a building life cycle.

The economy of GBs is the basic driving force and decision-making benchmark of its development. The ongoing debate over the economics of GBs seems to be where the potential financial saving can be made—in the initial investment in GBs, later operation costs, or reduced resource use. All of these could depend on individual cases. Surely this remains one of the interesting areas for further studies. Green Building Rating Systems are developed and applied by most countries all over the world as a guideline to achieve sustainable building construction goals. This paper summarizes 47 certification standards related to GBs in the world. LEED in the United States and BREEAM in the United Kingdom have the largest market shares. The certification expert mechanism guarantees the professional quality of consultants and project quality, but only in 16 certification standards. Other standards need to be enhanced in this regard. The importance of economic aspects of GBs was emphasized in much of the literature, but detailed analyses are limited.

This extensive survey suggests that most GB studies focus on certification standard analysis and comparison, technologies solutions in terms of energy performance, water efficiency, and indoor environmental quality. This paper provides useful recommendations from the technologies side, management side, and occupants side, finding that there is low participation among stakeholders, especially occupants participating in the development of GB in many countries. Mismatching technologies utilization due to lack of knowledge requires more consideration in future research. This paper proposes involving integrated management and exploring occupants' behavior and feedback to improve GB efficiency. Meanwhile, providing training and education in using GB technologies for occupants, as well as raising the awareness of local environmental issues, are expected in the future.

**Author Contributions:** Conceptualization, Y.Z. and H.W.; methodology, Y.Z.; validation, W.G., and F.W.; formal analysis, Y.Z. and H.W.; investigation, Y.Z.; resources, Y.Z. and H.W.; writing—original draft preparation, Y.Z.; writing—review and editing, W.G., F.W., N.Z., D.M.K., and X.Y.; visualization, Y.Z.; supervision, W.G.; project administration, Y.Z.

**Funding:** This research was funded by Grant-in-Aid for Scientific Research (C), grant number 17K06719.

**Conflicts of Interest:** The authors declare no conflict of interest.

## Appendix A

**Table A1.** Green Building Rating Systems (GBRSs) list in various countries.

| No. | Time Issued | Standard | Countries | Leading Organization | Full Name |
|---|---|---|---|---|---|
| 1 | 1990 | BREEAM | United Kingdom | Building Research Establishment Ltd. (BRE) | Building Research Establishment's Environmental Assessment Method |
| 2 | 1993 | BEPAC | Canada | The University of British Columbia | Building Environmental Performance Assessment Criteria |
| 3 | 1998 | LEED | United States | U.S. Green Building Council (USGBC) | Leadership in Energy and Environmental Design |
| 4 | 2002 | PromisE | Finland | VTT Technical Research Station | The Finnish Environmental Assessment and Classification System |
| 5 | 2010 | BEAM Plus | Hong Kong | Hong Kong Green Building Council and the BEAM Society Limited | Built Environmental Assessment Method |
| 6 | 1997 | EcoEffect | Sweden | The Royal Institute of Technology, Stockholm and the University of Gavle | ―― |
| 7 | 1998 | GBA/GBTool | Canada | International Initiative for a Sustainable Built Environment | The Green Building Assessment (GBA) |
| 8 | 1999 | NABERS/ABGR | Australia | The Office of Environment and Heritage (OEH) | National Australian Built Environment Rating System/Australian Building Greenhouse Rating system |
| 9 | 1999 | EEWH | China (Taiwan) | National Council for Sustainable Development under the Ministry of the Interior (MOI) | Green Building Labeling System |
| 10 | 1999 | Eco-Quantum | Netherlands | IVAM | ―― |
| 11 | 2000 | GG | Canada | ECD Energy and Environment Canada | Green Globes |
| 12 | 2000 | BEAT | Denmark | Danish Building Research Institute (SBI) | Building Evaluation Assessment Tool |
| 13 | 2000 | Ecoprofil | Norway | Norwegian Building Research Institute (SINTEF Byggforsk) | Ökoprofil |
| 14 | 2001 | CASBEE | Japan | Japan Sustainable Building Consortium (JSBC) | Comprehensive Assessment System for Building Environmental Efficiency |
| 15 | 2002 | CEPAS | Hong Kong | Building Department of Hong Kong Special Administrative Region of the People's Republic of China | Comprehensive Environmental Performance Assessment Scheme |
| 16 | 2002 | KGBC | Korea | Korea Green Building Council | Korea Green Building Certification System |
| 17 | 2003 | GS | Australia | Green Building Council Australia | Green Star |
| 18 | 2003 | TGBRS | India | The Energy and Resources Institute (TERI) | Teri Green Building Rating System |
| 19 | 2004 | GRIHA | India | The Energy and Resources Institute (TERI) | Green Rating for Integrated Habitat Assessment |
| 20 | 2005 | HQE | France | Cerway | Haute Qualite Environment |
| 21 | 2005 | Si-5281 | Israel | Standard Institute of Israel | Israel Standard 5281: Building with Reduced Environmental Impact |
| 22 | 2005 | GM | Singapore | Building and Construction Authority (BCA) | Green Mark |
| 23 | 2006 | LBC | America | International Living Future Institute | Living Building Challenge |
| 24 | 2006 | GPR | America | Built It Green | GreenPint Rated |
| 25 | 2006 | ASGB | China | Ministry of Housing and Urban-Rural Development of People's Republic of China | Assessment Standard for Green Building |
| 26 | 2006 | DGNB | Germany | The German Sustainable Building Council (Non-profit organization) | Deutscbe Gesellschaft Fur Nachhaltiges Bauen |
| 27 | 2006 | CSH | United Kingdom | Department for Communities and Local Government | Code for Sustainable Homes |
| 28 | 2007 | EPRS | Abu Dhabi | Abu Dhabi Urban Planning Council GBI | Estidama Pearl Rating System |

**Table A1.** *Cont.*

| No. | Time Issued | Standard | Countries | Leading Organization | Full Name |
|---|---|---|---|---|---|
| 30 | 2007 | SICES | Mexico | The Mexico Green Building Council (MGBC) | Sustainable Building Rating Tool/Sistema de Calificación de Edificación Sustentable |
| 31 | 2008 | NGBS | America | National Association of Home Builders (NAHB) | National Green Building Standard |
| 32 | 2008 | AQUA-HQE | Brasil | Vanzolini Foundation at the Polytechnic University of Sao Paulo | Alta Qualidade Ambientale |
| 33 | 2008 | LiderA | Portugal | Manuel Duate Pinheiro, Ph.D. | The Sistema de Acaliacao da Sustentabilidade (Certification System of Environmentally Sustainable Construction) |
| 34 | 2009 | ITACA Protocal | Italy | Institute for Innovation, Procurement Transparency and Compatibility Environmental-National Association of Regions and Autonomous Provinces (ITACA) | Protocollo Itaca |
| 35 | 2009 | GBI | Malaysia | Architectural Association of Malaysia(PAM) | Green Building Index |
| 36 | 2009 | BERDE | Philippine | Philippine Green Building Council (PHILGBC) | Building for Ecologically Responsive Design Excellence |
| 37 | 2009 | GSAS | Qatar | Gulf Organization for Research & Development | Global Sustainability Assessment System |
| 38 | 2009 | VERDE | Spain | Green Building Council España (GBCE) | Herramienta VERDE |
| 39 | 2010 | GPRS | Egypt | Egypt Green Building Council | The Green Pyramid Rating System Levels |
| 40 | 2010 | LOTUS | Vietnam | Vietnam Green Building Council (VGBC) | —— |
| 41 | 2010 | GREENSHIP | Indonesia | Green Building Council Indonesia | —— |
| 42 | 2010 | TREES | Thailand | Thai Green Building Institute | Thai's Rating of Energy and Environmental Sustainability |
| 43 | 2010 | BNB | GERMANY | the Federal Ministry of the Interior, Building and Community | Assessment System for Sustainable Building |
| 44 | 2012 | ARZ BRS | Lebanon | Lebanon Green Building Council (LGBC) | ARZ Building Rating System |
| 45 | 2013 | IGBC | India | Indian Green Building Council | Indian Green Building Council Rating system |
| 46 | 2014 | EDGE | America | International Finance Corporation −World bank group Green | Excellence in Design for Greater Efficiencies (EDGE) |
| 47 | 2014 | WELL | America | The International WELL Building Institute (IWBI) | —— |
| 48 | 2017 | CASA Colombia | Colombia | Consejo Colombiano de Construccion Sostenibe (CCCS) | —— |
| 49 | 2018 | CEDBIK-Konut | Turkey | Turkey Green Building Council | Cevre Dostu Yesil Binalar Dernegi |

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
