# Peer review of "A Survey of the Status and Challenges of Green Building Development in Various Countries"

_sustainability, doi:10.3390/su11195385_

Round 1

Reviewer 1 Report

Interesting and exhaustive review article, which analyses in detail the state of the art of the Green Building Development in Various Countries. In the paper, 47 worldwide green building standards have been analysed and the research results and lessons learned from green building projects in different countries are discussed. The topic is of interest for the scientific community and specially for policy makers and construction design companies. The writing and structure of the manuscript is correct, although the following minor changes are needed in order to improve quality of the paper:

This reviewer miss (within section 3: Background and definition) a mention of the initiatives that have been taken and are being taken from the academic and scientific field in order to mitigate the effects of climate change associated with the activity of the construction sector. Specifically, I miss to mention research that have developed more sustainable construction systems, through the replacement of conventional concrete or steel construction elements with timber elements, such as:

García, H., Zubizarreta, M., Cuadrado, J., & Osa, J. (2019). Sustainability Improvement in the Design of Lightweight Roofs: A New Prototype of Hybrid Steel and Wood Purlins. Sustainability, 11(1), 39.

I also miss mentioning the development of simple and rapid sustainability assessment models specific to timber structures and buildings, whose objective is to design and project timber buildings in the most sustainable way possible, with the ultimate goal of reducing the impact that the construction sector activity has in the environment, such as:

Zubizarreta, M., Cuadrado, J., Orbe, A., & García, H. (2019). Modeling the environmental sustainability of timber structures: A case study. Environmental Impact Assessment Review, 78, 106286.

Author Response

Dear Reviewers and Editors,

We are very grateful to the referee and Area Editor of the paper for their critical reading of the manuscript and valuable recommendation for our further improvements. We have checked the manuscript and revised it according to the comments.

We have added the suggested content in section 5.1 technologies application -the fifth paragraph of building material with citation of the mentioned reference paper.

Reviewer 2 Report

A review of literature on various green building assessment schemes is presented and the application of green building schemes in various countries is discussed.

It is stated that a certain percentage of buildings are green buildings (for example in China). The paper does not discuss the quality required by the various green building schemes. A CGBL indicates a different level of quality, than a DGNB certification. This aspect is missing in the discussion of the paper.

In parts of the paper very specialized examples are given. These example appear not to be appropriate and the selection seems to be random.- The authors should consider to explain the aspects without these example - they have no value for the paper. 

please see the notes in the attachment.

Author Response

Dear Reviewers and Editors,

We are very grateful to the referee and Area Editor of the paper for their critical reading of the manuscript and valuable recommendation for our further improvements. We have checked the manuscript and revised it according to the comments.
